# Association between genetically proxied PCSK9 inhibition and prostate cancer risk: A Mendelian randomisation study

**Si Fang**[1,2]*, **James Yarmolinsky**[1,2], **Dipender Gill**[3,4], **Caroline J. Bull**[1,2,5,6], **Claire M. Perks**[6], **the PRACTICAL Consortium**[¶], **George Davey Smith**[1,2], **Tom R. Gaunt**[1,2], **Tom G. Richardson**[1,2]*

1 Population Health Sciences, Bristol Medical School, University of Bristol, Bristol, United Kingdom, 2 Medical Research Council (MRC) Integrative Epidemiology Unit (IEU), University of Bristol, Bristol, United Kingdom, 3 Chief Scientific Advisor Office, Research and Early Development, Novo Nordisk, Copenhagen, Denmark, 4 Department of Epidemiology and Biostatistics, School of Public Health, Imperial College London, London, United Kingdom, 5 Bristol Renal, Bristol Heart Institute, Translational Health Sciences, Bristol Medical School, University of Bristol, Bristol, United Kingdom, 6 IGF & Metabolic Endocrinology Group, Translational Health Sciences, Bristol Medical School, Learning & Research Building, Southmead Hospital, Bristol, United Kingdom

¶ Membership of the PRACTICAL consortium is provided in S1 File.
* Si.Fang@bristol.ac.uk (SF); Tom.G.Richardson@bristol.ac.uk (TGR)

**Data Availability Statement:** Several summary statistics of genome wide association studies (GWAS) used in this study are publicly available on

## Abstract

### Background

Prostate cancer (PrCa) is the second most prevalent malignancy in men worldwide. Observational studies have linked the use of low-density lipoprotein cholesterol (LDL-c) lowering therapies with reduced risk of PrCa, which may potentially be attributable to confounding factors. In this study, we performed a drug target Mendelian randomisation (MR) analysis to evaluate the association of genetically proxied inhibition of LDL-c-lowering drug targets on risk of PrCa.

### Methods and findings

Single-nucleotide polymorphisms (SNPs) associated with LDL-c ($P < 5 \times 10^{-8}$) from the Global Lipids Genetics Consortium genome-wide association study (GWAS) ($N = 1,320,016$) and located in and around the *HMGCR*, *NPC1L1*, and *PCSK9* genes were used to proxy the therapeutic inhibition of these targets. Summary-level data regarding the risk of total, advanced, and early-onset PrCa were obtained from the PRACTICAL consortium. Validation analyses were performed using genetic instruments from an LDL-c GWAS conducted on male UK Biobank participants of European ancestry ($N = 201,678$), as well as instruments selected based on liver-derived gene expression and circulation plasma levels of targets. We also investigated whether putative mediators may play a role in findings for traits previously implicated in PrCa risk (i.e., lipoprotein a (Lp(a)), body mass index (BMI), and testosterone).

Applying two-sample MR using the inverse-variance weighted approach provided strong evidence supporting an effect of genetically proxied inhibition of PCSK9 (equivalent to a

the MRC Integrative Epidemiology Unit (IEU) Open GWAS project (https://gwas.mrcieu.ac.uk/), including LDL cholesterol GWAS from the Global Lipids Genetics (GLGC) Consortium, overall prostate cancer risk GWAS from the Prostate Cancer Association Group to Investigate Cancer Associated Alterations in the Genome (PRACTICAL) consortium, Locke et al. GWAS on body mass index, GWAS on lipoprotein A, male-stratified total and bioavailable testosterone from the Neale lab, as well as the newly performed male-stratified GWAS on LDL cholesterol on UK Biobank participants. Pulit et al. GWAS summary statistics on body mass index is available from https://zenodo.org/record/1251813#.YlhVzZPMJhE. Liver-derived gene expression eQTL data is available from the GTEx project via https://www.gtexportal.org/home/. The data used for the analyses described in this manuscript were obtained from the GTEx Portal on 12/12/2021. PCSK9 plasma pQTL data is available from https://www.decode.com/summarydata/. Summary statistics of GWAS on advanced and early onset prostate cancer risks are under restricted access. They are available from the PRACTICAL consortium upon application (contact: PRACTICAL@icr.ac.uk).

**Funding:** This research was conducted in the Medical Research Council (MRC) Integrative Epidemiology Unit at the University of Bristol (MC_UU_00011/1 to GDS, MC_UU_00011/4 to TRG) (https://www.ukri.org/councils/mrc/). SF is supported by a Wellcome Trust PhD studentship in Molecular, Genetic and Lifecourse Epidemiology (108902/Z/15/Z) (https://wellcome.org/). JY is supported by a Cancer Research UK Population Research Postdoctoral Fellowship (C68933/A28534) (https://www.cancerresearchuk.org/). CJB is supported by the World Cancer Research Fund (WCRF UK) (https://www.wcrf.org/), as part of the World Cancer Research Fund International grant program (IIG_2019_2009). TRG and GDS conduct research at the NIHR Biomedical Research Centre at the University Hospitals Bristol NHS Foundation Trust and the University of Bristol. The views expressed in this publication are those of the author(s) and not necessarily those of the NHS, the National Institute for Health Research or the Department of Health. The funders had no role in study design, data collection and analysis, decision to publish, or preparation of the manuscript.

**Competing interests:** I have read the journal's policy and the authors of this manuscript have the following competing interests: DG is employed part-time by Novo Nordisk outside of this work. TGR is an employee of GlaxoSmithKline outside of this research. TRG and GDS receives funding from

standard deviation (SD) reduction in LDL-c) on lower risk of total PrCa (odds ratio (OR) = 0.85, 95% confidence interval (CI) = 0.76 to 0.96, $P = 9.15 \times 10^{-3}$) and early-onset PrCa (OR = 0.70, 95% CI = 0.52 to 0.95, $P = 0.023$). Genetically proxied HMGCR inhibition provided a similar central effect estimate on PrCa risk, although with a wider 95% CI (OR = 0.83, 95% CI = 0.62 to 1.13, $P = 0.244$), whereas genetically proxied NPC1L1 inhibition had an effect on higher PrCa risk with a 95% CI that likewise included the null (OR = 1.34, 95% CI = 0.87 to 2.04, $P = 0.180$). Analyses using male-stratified instruments provided consistent results.

Secondary MR analyses supported a genetically proxied effect of liver-specific *PCSK9* expression (OR = 0.90 per SD reduction in *PCSK9* expression, 95% CI = 0.86 to 0.95, $P = 5.50 \times 10^{-5}$) and circulating plasma levels of PCSK9 (OR = 0.93 per SD reduction in PCSK9 protein levels, 95% CI = 0.87 to 0.997, $P = 0.04$) on PrCa risk. Colocalization analyses identified strong evidence (posterior probability (PPA) = 81.3%) of a shared genetic variant (rs553741) between liver-derived *PCSK9* expression and PrCa risk, whereas weak evidence was found for *HMGCR* (PPA = 0.33%) and *NPC1L1* expression (PPA = 0.38%). Moreover, genetically proxied PCSK9 inhibition was strongly associated with Lp(a) levels (Beta = −0.08, 95% CI = −0.12 to −0.05, $P = 1.00 \times 10^{-5}$), but not BMI or testosterone, indicating a possible role for Lp(a) in the biological mechanism underlying the association between PCSK9 and PrCa. Notably, we emphasise that our estimates are based on a life-long exposure that makes direct comparisons with trial results challenging.

## Conclusions

Our study supports a strong association between genetically proxied inhibition of PCSK9 and a lower risk of total and early-onset PrCa, potentially through an alternative mechanism other than the on-target effect on LDL-c. Further evidence from clinical studies is needed to confirm this finding as well as the putative mediatory role of Lp(a).

## Author summary

### Why was this study done?

- Prostate cancer is the second most diagnosed malignancy in men globally.

- Previous studies have provided conflicting evidence regarding a relationship between elevated low-density lipoprotein (LDL) cholesterol and prostate cancer risk.

- The aim of this study was to examine the association between genetically proxied inhibition of lipid-lowering drug targets (i.e., PCSK9, NPC1L1, HMGCR) and prostate cancer using evidence from multiple datasets and analytical methods.

### What did the researchers do and find?

- Using genetic variants associated with LDL cholesterol, liver-derived gene expression, and plasma protein levels, the researchers applied drug target Mendelian randomisation

Biogen for unrelated research. GDS is an Academic Editor on PLOS Medicine's editorial board. All other co-authors declare no conflict of interest.

**Abbreviations:** BMI, body mass index; CHD, coronary heart disease; CI, confidence interval; CLPP, colocalization posterior probability; GLGC, Global Lipids Genetics Consortium; GTEx, Genotype-Tissue Expression; GWAS, genome-wide association study; IVW, inverse-variance weighted; LD, linkage disequilibrium; LDL, low-density lipoprotein; LDL-c, LDL cholesterol; MR, Mendelian randomization; OR, odds ratio; PRACTICAL, Prostate Cancer Association Group to Investigate Cancer Associated Alterations in the Genome; SD, standard deviation; SNP, single-nucleotide polymorphism.

(MR) and colocalization to examine the association between lipid-lowering drug targets and the risk of overall, early-onset, and advanced prostate cancer. Additional MR analyses were conducted to explore putative mediators of drug effects.

- This study provided evidence of an association between genetically proxied PCSK9 inhibition and lower risk of overall and early-onset prostate cancer supported by both MR and colocalization approaches.

- Follow-up analyses of genetically proxied PCSK9 inhibition highlighted a potential mediatory role for Lp(a) along the causal pathway to lower prostate cancer risk.

## What do these findings mean?

- PCSK9 inhibition may be involved in biological mechanisms that reduce the risk of overall and early-onset prostate cancer, potentially through the regulation of Lp(a).

- However, functional validation is necessary to confirm these findings, as well as future research to further evaluate the relationship between lipid-lowering drug targets and advanced prostate cancer risk.

## Introduction

Prostate cancer is the second most commonly diagnosed malignancy in men globally with over 1.4 million new cases in 2020 [1]. Findings from the literature have provided conflicting evidence of a relationship between elevated low-density lipoprotein (LDL) cholesterol and prostate cancer risk. For example, preclinical studies have suggested that high levels of extracellular LDL cholesterol (LDL-c) may promote the proliferation of prostate cancer cells [2,3]. Conversely, previous observational studies [4–8] have typically found limited evidence of an association between the levels of LDL-c and overall risk of prostate cancer, although some have reported that LDL-c lowering medications may reduce the risk of prostate cancer incidence [6,9–12]. Taken together, these findings suggest that although LDL-c may not directly contribute towards prostate tumorigenesis, biological pathways that regulate the biosynthesis and metabolism of LDL-c may influence prostate cancer risk through alternate mechanisms.

The use of human genetics to evaluate the efficacy and safety profiles of therapeutic targets is becoming increasingly popular in drug development, with recent evidence suggesting that targets with the support of genetics are approximately twice as likely to successfully make it to market [13]. Furthermore, the wealth of readily accessible data from genome-wide association studies (GWAS) means that these types of evaluations are typically inexpensive and quick to undertake. An approach to investigate genetic support for a target is Mendelian randomisation (MR) [14–16], a causal inference technique that harnesses randomly segregated genetic variants within a population as instrumental variables to proxy the perturbation of therapeutic targets [17,18].

The application of MR to examine the genetically proxied effects of drug targets (referred to as "drug target MR") has demonstrated the validity of this approach to corroborate findings from clinical trials as long as the underlying assumptions are met [19]. For example, the efficacy of lipid-lowering drug targets in reducing risk of coronary artery disease has been shown

by previous MR studies, for therapies such as statins (which target HMG-CoA reductase (*HMGCR*)), proprotein convertase subtilisin/kexin type 9 (*PCSK9*) inhibitors and Ezetimibe (which targets Niemann–Pick C1-Like 1 (*NPC1L1*)) [20,21]. Moreover, drug target MR analyses have provided evidence of adverse effects reported in trials, such as the effect of statins on increased risk of type 2 diabetes, as well as highlighting potential additional indications [22]. This approach has also been applied to suggest that statins may provide additional benefit towards the lowering of ovarian cancer risk [23], as well as a recent study by Sun and colleagues that investigated evidence of association between genetically proxied lipid-lowering drugs with breast and prostate cancer [12]. However, this study did not triangulate findings from multiple sources of data (such as molecular traits) or evaluate the robustness of MR estimates using different research methods, such as genetic colocalization.

In this study, we applied drug target MR to investigate the association between genetically proxied LDL-c lowering medications and risk of total, advanced, and early-onset prostate cancer using data from the Prostate Cancer Association Group to Investigate Cancer Associated Alterations in the Genome (PRACTICAL) consortium [24]. Genetic instruments were oriented to proxy the effect of statins, PCSK9 inhibitors, and Ezetimibe on lowering circulating LDL-c. The associations of genetically proxied LDL receptor (LDLR) mediated LDL-c and the genetically proxied levels of circulating LDL-c on prostate cancer outcomes were also examined to assess whether evidence of the findings for inhibiting drug targets may be generalisable to the lowering of LDL-c. Moreover, analyses using genetic variants to investigate the levels of circulating proteins and liver tissue-derived gene expression for targets were performed as sensitivity analyses and to triangulate evidence [25]. Effect estimates on advanced prostate cancer and early-onset prostate cancer were also investigated. Finally, we evaluated the genetically proxied effects of LDL-c-lowering drugs on several traits previously implicated to play a role in prostate cancer risk (body mass index (BMI) [26–28], lipoprotein A (Lp(a)) [8], and testosterone [29,30]) to discern whether they may reside along the pathways between therapeutic targets and risk of prostate cancer.

## Methods

This study is reported as per the Strengthening the Reporting of Observational Studies in Epidemiology (STROBE) guideline, specific for Mendelian randomisation [31] (S1 STROBE Checklist). An overview of the study can be found in Fig 1.

### Ethics statement

Our work involved the previously collected genetic sequencing and phenotype data of human participants in the UK Biobank cohort study. The North West Multi-centre Research Ethics Committee (MREC) gave ethical approval for the UK Biobank.

### Data sources

In our primary analysis, two-sample MR was applied to study the association of genetically proxied therapeutic inhibition for lipid-lowering drug targets PCSK9, NPC1L1, and HMGCR, as well as genetically proxied levels of LDLR and overall LDL-c, on the risk of prostate cancer.

Genetic instruments for drug targets were extracted from the latest summary statistics from the Global Lipids Genetics Consortium (GLGC) GWAS on LDL-c levels ($n = 1,320,016$) [32] from the online data repository (http://csg.sph.umich.edu/willer/public/glgc-lipids2021/). Final instruments for the 3 lipid-lowering drug targets (and genetically proxied LDLR) were genetic variants robustly associated with LDL-c (based on $P < 5 \times 10^{-8}$ and a pairwise linkage disequilibrium (LD) $r^2 < 0.1$ using a reference panel consisting of individuals of European

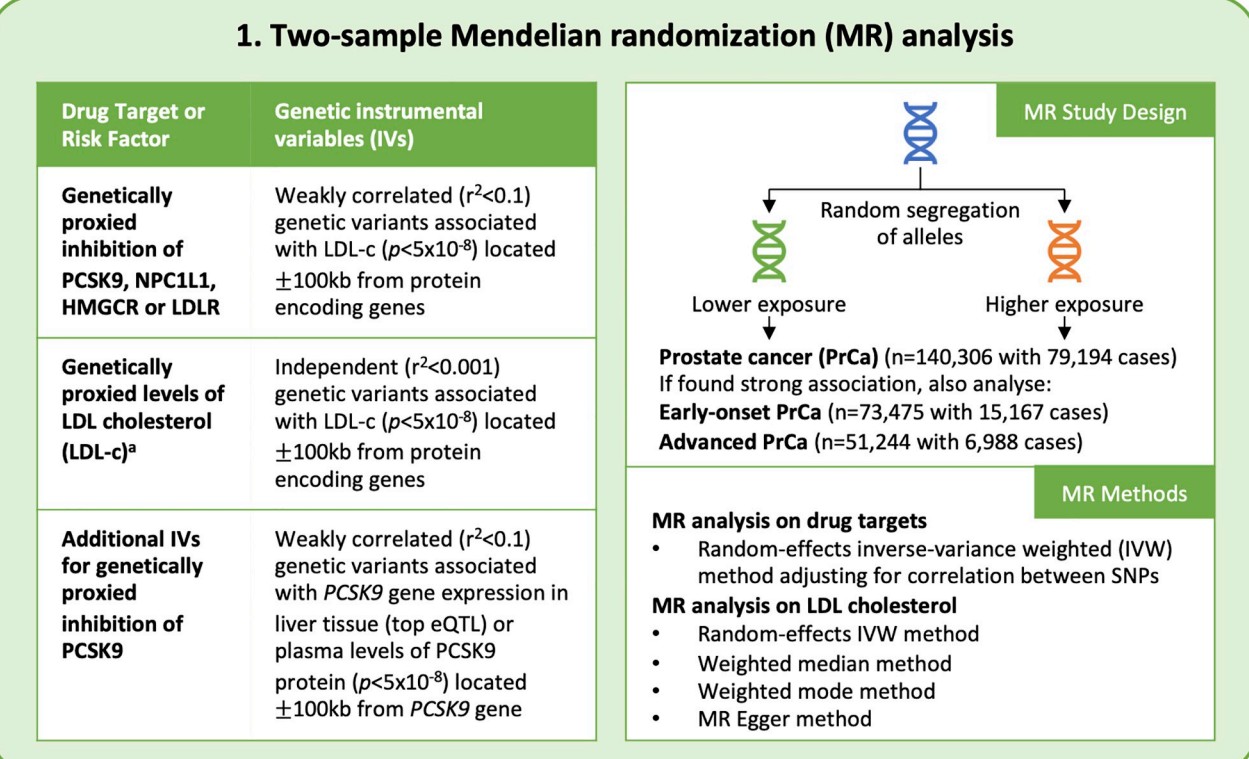

## 1. Two-sample Mendelian randomization (MR) analysis

| Drug Target or Risk Factor | Genetic instrumental variables (IVs) |
|---|---|
| **Genetically proxied inhibition of PCSK9, NPC1L1, HMGCR or LDLR** | Weakly correlated ($r^2$<0.1) genetic variants associated with LDL-c ($p$<5x10$^{-8}$) located ±100kb from protein encoding genes |
| **Genetically proxied levels of LDL cholesterol (LDL-c)[a]** | Independent ($r^2$<0.001) genetic variants associated with LDL-c ($p$<5x10$^{-8}$) located ±100kb from protein encoding genes |
| **Additional IVs for genetically proxied inhibition of PCSK9** | Weakly correlated ($r^2$<0.1) genetic variants associated with *PCSK9* gene expression in liver tissue (top eQTL) or plasma levels of PCSK9 protein ($p$<5x10$^{-8}$) located ±100kb from *PCSK9* gene |

**MR Study Design**

Random segregation of alleles

Lower exposure — Higher exposure

**Prostate cancer (PrCa)** (n=140,306 with 79,194 cases)
If found strong association, also analyse:
**Early-onset PrCa** (n=73,475 with 15,167 cases)
**Advanced PrCa** (n=51,244 with 6,988 cases)

**MR Methods**

**MR analysis on drug targets**
• Random-effects inverse-variance weighted (IVW) method adjusting for correlation between SNPs

**MR analysis on LDL cholesterol**
• Random-effects IVW method
• Weighted median method
• Weighted mode method
• MR Egger method

## 2. Co-localization between eQTL and total prostate cancer risk

**Colocalization Datasets**
• GWAS of *PCSK9* gene expression in the liver
• GWAS of overall prostate cancer risk

**Colocalization Methods**
• Visualization with LocusZoom plots;
• Quantitative appraisal using *coloc* and eCAVIAR

## 3. Contrasting the associations between drug targets and PrCa risk

**Risk Factor of PrCa = Potential Mediators**

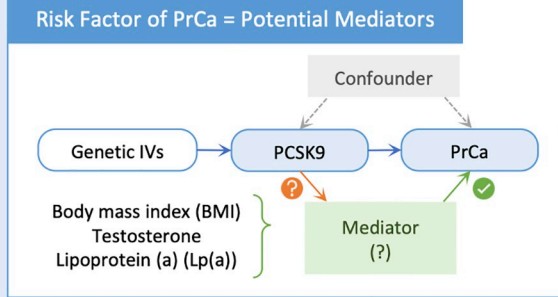

Confounder

Genetic IVs → PCSK9 → PrCa

Body mass index (BMI)
Testosterone
Lipoprotein (a) (Lp(a))

Mediator (?)

**MR Methods**

**Drug targets – mediators association:**
• Random-effects IVW method adjusting for correlation between SNPs (LD matrix)

**Replication of Lp(a) – PrCa association[b]:**
• Univariable MR using 15 genome-wide significant SNPs associated with Lp(a) ($r^2$<0.001, $p$<5x10$^{-8}$) (random-effects IVW, weighted median) or a variant on *LPA* gene (Wald ratio)
• Multivariable MR adjusting for LDL-c levels

**Fig 1. Study overview.** [a]The LDL-c variants were oriented to be in the LDL-c lowering direction, i.e., effect estimates in Mendelian randomisation analysis are per 1 SD lowering of LDL-c. [b]The original analysis was conducted by Ioannidou and colleagues [8]. LDL-c, LDL cholesterol; MR, Mendelian randomization; SD, standard deviation.

ancestry from the 1,000 Genomes Project Consortium [33]) and located within 100 kb around *PCSK9* (Entrez Gene: 255738), *NPC1L1* (Entrez Gene: 29881), *HMGCR* (Entrez Gene: 3156), and *LDLR* (Entrez Gene: 3949), respectively. Additionally, genome-wide variants associated with LDL-c were used as instrumental variables for overall levels of LDL-c (based on $P < 5 \times 10^{-8}$ and $r^2 < 0.001$).

Given that genetic estimates on prostate cancer risk were derived using a male-only population, we repeated all analyses using male-stratified instruments identified by conducting a GWAS of LDL-c (ID: 30780) on 201,678 male participants of European ancestry in the UK Biobank study [34]. The levels of LDL-c were standardised to have a mean of 0 and SD of 1 prior to analysis. BOLT-LMM was implemented to conduct the GWAS analysis with adjustment for age and genotyping chip [35]. Further details of the GWAS analysis pipeline has been described previously [36,37]. After quality control and imputation, the GWAS results were clumped with 2 sets of different *P*-value and LD $r^2$ thresholds as described above, i.e., (1) $P < 5 \times 10^{-8}$ and pairwise $r^2 < 0.1$, or (2) $P < 5 \times 10^{-8}$ and $r^2 < 0.001$.

For prostate cancer outcomes, summary statistics were obtained from GWAS meta-analyses on the risks of overall prostate cancer (*n* = 140,306 men including 79,194 cases and 61,112 controls), as well as 2 stratified meta-analysis, including: (1) the risk of early-onset prostate cancer (*n* = 51,244 including 6,988 prostate cancer cases diagnosed on or before age 55 years and 44,256 non-prostate cancer controls); (2) the risk of advanced prostate cancer (*n* = 73,475 including 15,167 advanced prostate cancer cases and 58,308 non-prostate cancer controls), all from the PRACTICAL consortium in which the majority of studies includes cases and controls without matching by clinical features [24]. Advanced prostate cancer cases include individuals with either metastatic prostate cancer, a Gleason score of 8 or higher, a prostate-specific antigen level greater than 100 ng/mL, or prostate cancer-related death [24]. All individuals involved in the prostate cancer GWAS analyses are of European ancestry.

We also used various datasets as part of our secondary analyses in this paper. Full details of these can be found in S1 Supplementary Note. A summary of all GWASs involved in this study is in S1 Table.

## Statistical analysis

**Two-sample Mendelian randomisation.**  In the primary analysis, two-sample MR was applied to investigate the associations of genetically proxied inhibition of lipid-lowering drug targets and overall LDL-c on the risk of prostate cancer. All analysis was performed using the *TwoSampleMR* R package (v0.5.6, https://github.com/mrcieu/TwoSampleMR). The application of MR must satisfy 3 key assumptions, including (1) the genetic instrumental variable are strongly associated with the exposure of interest; (2) they are associated with the outcome only through the exposure; and (3) the exposure and outcome does not have a shared cause [17,18]. The first assumption is the only testable assumption and could be assessed using instrument strength. The instrument strength (F statistics) for LDL-c-lowering drug targets and risk factors examined in this study were calculated using a formula previously described by Bowden and colleagues [38].

In the primary drug target MR analysis, effect estimates for genetically proxied inhibition of PCSK9, NPC1L1, and HMGCR were derived using the random-effects inverse-variance weighted (IVW) model [39]. Considering the weak LD between genetic variants ($r^2 < 0.1$) used as instrumental variables, IVW analyses were adjusted for LD matrices between instruments based on the same reference panel as above to ensure they were independent of one and other [40]. Iterative leave-one-out analyses were conducted for PCSK9 to identify the presence of any single variants that may be driving identified effects on the outcome. In validation

analysis using UKB male-stratified genetic instruments, the effects from inhibiting PCSK9 and HMGCR were analysed using IVW accounting for genetic correlations, whereas the effects from inhibition of NPC1L1 was estimated using Wald ratio based on rs2073547.

Next, we performed two-sample MR using the random-effects IVW method to investigate whether the association between genetically proxied LDL-c-lowering drug targets and total prostate cancer risk may be attributed to the inhibition of LDLR or due to overall changes in LDL-c. When analysing the effects of LDL-c levels, the weighted median model (which allows up to half of the included SNPs to be pleiotropic and is less influenced by outliers) [41], the weighted mode model (which assumes that the most common effect is consistent with the true causal effect) [42], and the MR-Egger model (which provides an estimate of association magnitude allowing all SNPs to be pleiotropic) [43] were used as sensitivity analysis.

As a secondary analysis, two-sample MR were performed using PCSK9 *cis*-eQTL and *cis*-pQTL to further examine results identified in the primary analyses. We firstly estimated the association between genetically proxied plasma levels of PCSK9 and LDL-c using the random-effects IVW methods accounting for LD between genetic variants. A Wald ratio was calculated to estimate the association between genetically proxied *PCSK9* expression (instrumented using a single *cis*-eQTL) and prostate cancer outcomes. We applied the random-effects IVW method accounting for correlation structure between genetic variants to examine the associations between genetically proxied plasma PCSK9 (instrumented using *cis*-pQTLs) and prostate cancer outcomes. The pairwise LD correlation $r^2$ between eQTL and pQTLs were calculated using LDmatrix from LDlink [44] based on the reference panel consisting of Utah Residents from North and West Europe (CEU) individuals. The PCSK9 *cis*-pQTL MR analysis was repeated using conditionally independent *cis*-pQTLs (based on the same Icelandic population they were derived from) using random-effect IVW method without the adjustment of correlation between SNPs.

In addition, LDL-c-associated genetic variants at the *PCSK9* locus were functionally annotated using Ensembl Variant Effect Predictor (VEP) [45]. Regulatory pathway data for all targets analysed in this study were queried using the STRING (v11) database based on experimentally determined data [46].

**Co-localization between eQTL and total prostate cancer risk.** Given that single SNP MR analyses can be prone to high false discovery rates due to LD between the instrument and proximal variants, we conducted co-localization analyses to identify evidence of shared causal variants between liver-derived gene expression and risk of total prostate cancer. We constructed LocusZoom plots using *gassocplot2* R package (https://github.com/jrs95/gassocplot2) to visualise the genetic variants associated with liver-derived *PCSK9*, *HMGCR*, and *NPC1L1* gene expression (GTEx v8) at each of their corresponding loci and variants associated with risk of prostate cancer. The *coloc* (v5.1.0) [47] and eCAVIAR (v2.2) [48] approaches were applied to formally appraise evidence using the genetic correlation matrix generated using European individuals from the 1,000 genome reference panel [33]. Colocalization using the *coloc* method quantified the probability of a shared genetic variant between liver tissue-derived gene expression and prostate cancer (H4) for all 3 genes, based on genetic variants within 300 kb up- and downstream the lead *cis*-eQTLs. eCAVIAR was further applied to quantify the colocalization posterior probability (CLPP) between *PCSK9* gene expression and prostate cancer, and the cut-off of CLPP to indicate evidence of a shared causal variant is >0.01 as proposed by the authors of this approach [48]. In addition, we estimated the genome-wide genetic correlation between LDL-c and prostate cancer using LD Score regression [49], as well as local genetic correlation between the 2 traits at the *PCSK9* locus using LAVA [50].

**Contrasting the genetically proxied associations between lipid-lowering drug targets and risk factors of prostate cancer.** To examine whether the associations between

genetically proxied inhibition of drug targets and prostate cancer risk could be attributed to a potential mediatory pathway involving changes in BMI, Lp(a), or testosterone, we performed two-sample MR to investigate effects from drug targets on those risk factors as a sensitivity analysis using the same methods as above. The associations between genetic variants and BMI were extracted (1) from the Pulit and colleagues GWAS meta-analysis on BMI ($n = 806,834$) [51] when using GLGC variants; and (2) from the Locke and colleagues GWAS on BMI ($n = 339,224$) [52] when analysed using UKB male-stratified genetic variants to avoid sample overlap. The associations between genetic instruments and Lp(a) were extracted from a GWAS on inverse rank normalised levels of Lp(a) in UKB participants (http://www.nealelab.is/uk-biobank/) from the IEU Open GWAS project. For BMI and Lp(a), replication analyses were conducted using summary statistics of male-stratified GWAS from the same cohorts mentioned above ($n = 374,756$ men in Pulit and colleagues GWAS on BMI, $n = 152,893$ men in Locke and colleagues GWAS on BMI, and $n = 167,020$ men in Neale lab UKB GWAS on inverse rank normalised levels of Lp(a)). Moreover, the associations between genetic variants and testosterone levels were from male-stratified GWAS on total and bioavailable testosterone levels ($n = 199,569$ and $184,205$ men, respectively) in the UK Biobank accessed through the IEU Open GWAS project. Random-effects IVW models were used with adjustment for the LD between instruments as above. For drug targets instrumented using a single genetic variant, Wald ratio estimates were used to evaluate their genetically proxied associations with the outcome.

In addition, for potential risk factors of prostate cancer found to be associated with genetically proxied PCSK9 inhibition, we conducted MR to further examine their associations with prostate cancer risk. This includes univariable and multivariable MR to replicate the previously published findings on Lp(a) and prostate cancer. Univariable MR was performed (1) using IVW and weighted median model with 15 genome-wide significant genetic variants associated with Lp(a) as genetic instruments; and (2) using Wald ratio estimate with the *cis*-acting variant associated with Lp(a) on LPA gene as the genetic instrument. Multivariable MR was performed using the *cis*-acting variant for Lp(a) together with genome-wide significant variants associated with LDL-c from the latest GLGC GWAS [32].

## Results

### Genetic variant selection

In total, 28 genetic variants were used to proxy the inhibition of PCSK9, 4 for NPC1L1, 13 for HMGCR, 36 for the levels of LDLR-mediated LDL-c, and 424 for the levels of total LDL-c identified using the latest GLGC GWAS. Details of these genetic variants are in S2 Table. Male-stratified genetic instruments for each of these exposures are presented in S3 Table. Details of the *cis*-acting protein quantitative trait loci (*cis*-pQTL) for plasma levels of PCSK9 protein and liver-derived *cis*-acting expression quantitative trait loci (*cis*-eQTL) data for *PCSK9* gene are listed in S4 Table. Functional annotations of PCSK9 variants are presented in S5 Table. Regulatory pathways between the 4 protein targets are presented in S1 Fig. The F statistics of genetic instruments for all drug targets and risk factors assessed in this study, including the eQTL and pQTLs, ranged from 26.9 to 629.7, suggesting that the results are unlikely to be biased due to weak instruments [53].

### Mendelian randomisation analysis of lipid-lowering therapies and prostate cancer risk

We firstly applied drug target MR to investigate the association of genetically proxied lipid-lowering drug targets (HMGCR, PCSK9, and NPC1L1) with overall prostate cancer risk (Fig 2

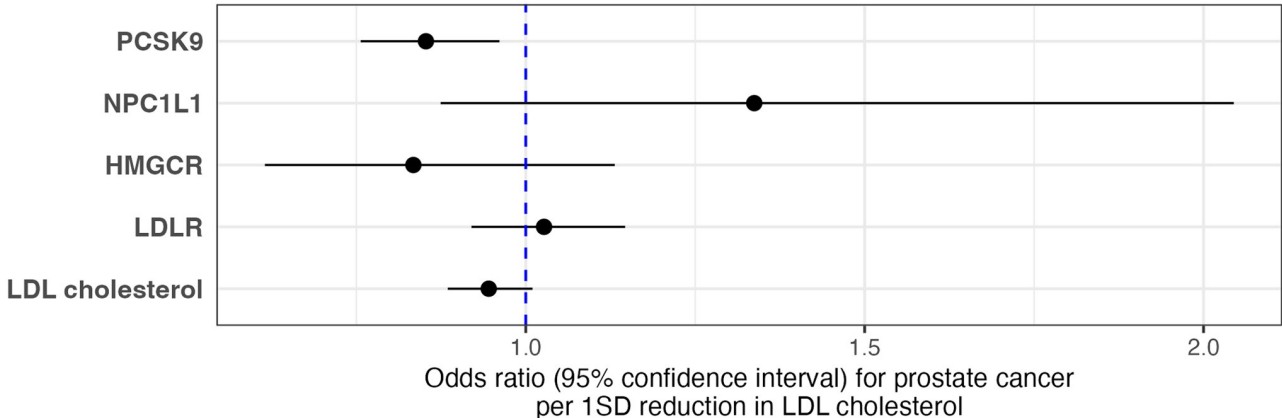

**Fig 2. Results from MR analyses to estimate the effect of lipid-lowering therapies and risk factors on overall prostate cancer risk.** Effect estimates are odds ratios for prostate cancer per 1 SD reduction in LDL-c proxied using genetic instruments identified from the GLGC. In total, 28 genetic variants were used to proxy the inhibition of PCSK9, 4 for NPC1L1, 13 for HMGCR, 36 for the levels of LDLR-mediated LDL-c, and 424 for the levels of total LDL-c identified using the latest GLGC GWAS. F-statistics for the exposures ranges from 221.5 to 629.7. Detailed results can be found in S6 Table. GLGC, Global Lipids Genetics Consortium; LDL-c, LDL cholesterol; MR, Mendelian randomization; SD, standard deviation.

and S6 Table). Genetically proxied inhibition of PCSK9 was strongly associated with a lower risk of developing prostate cancer (IVW MR odds ratio (OR) = 0.85, 95% confidence interval (95% CI) = 0.76 to 0.96, $P$ = 0.009, per standard deviation (SD) reduction in LDL-c). Leave-one-out analyses provided consistent evidence of an association between genetically proxied PCSK9 inhibition and risk of prostate cancer, suggesting that the overall estimate was not driven by a single influential variant (S2 Fig and S7 Table). Genetically proxied inhibition of HMGCR provided evidence of a similar magnitude of effect on overall prostate cancer as PCSK9, although the 95% CI included the null (OR = 0.83, 95% CI = 0.62 to 1.13, $P$ = 0.244). Genetically proxied inhibition of NPC1L1 was associated with higher overall prostate cancer risk which likewise had a 95% CI that included the null (OR = 1.34, 95% CI = 0.87 to 2.04, $P$ = 0.180). MR analysis on the risk of early-onset and advanced prostate cancer identified strong evidence of an association between genetically proxied inhibition of PCSK9 and early-onset disease (OR = 0.70, 95% CI = 0.52 to 0.95, $P$ = 0.023), but weaker evidence of association with advanced prostate cancer (OR = 0.91, 95% CI = 0.74 to 1.12, $P$ = 0.381) (S6 Table). Similar findings were observed using male-stratified genetic instruments identified using the UKB data (S6 Table).

For comparative purposes, we investigated the effect of circulating LDL-c on overall prostate cancer risk by selecting genetic instruments at the *LDLR* locus as well as those associated with LDL-c across the genome. There was minimal evidence to suggest an effect of genetically proxied LDLR mediated LDL-c levels on overall prostate cancer risk (OR = 1.04, 95% CI = 0.96 to 1.12, $P$ = 0.385, per SD reduction in LDL-c). Additionally, a set of 424 SNPs used as a genetic instrument for circulating LDL-c was weakly associated with prostate cancer risk (OR = 0.95, 95% CI = 0.88 to 1.01, $P$ = 0.096) with similar effect estimates to those found for HMGCR inhibition (heterogeneity $P$ = 0.432). The analysis for (overall) prostate cancer risk was repeated with similar conclusions using a male-specific 104 SNP LDL-c instrument derived in males only from the UKB (OR = 0.95, 95% CI = 0.89 to 1.01, $P$ = 0.101). MR-Egger, weighted median, and weighted mode estimates were comparable to the IVW results (S7 Table).

## Triangulation of evidence using data on circulating protein levels and liver-derived gene expression

To further examine the association between PCSK9 and prostate cancer, two-sample MR analyses were performed using *cis*-acting pQTLs to instrument inhibition of circulating PCSK9 protein levels (S8 Table). These *cis*-pQTLs are strongly associated with LDL-c levels (Beta = −0.54, 95% CI = −0.59 to −0.49, $P = 3.81 \times 10^{-90}$, SD change in LDL-c per SD reduction in PCSK9 levels). MR results provided evidence of an effect of lower levels of circulating PCSK9 protein on overall prostate cancer (OR = 0.93, 95% CI = 0.87 to 0.997, $P = 0.040$, per SD reduction in plasma PCSK9 levels), and early-onset prostate cancer (OR = 0.86, 95% CI = 0.74 to 0.98, $P = 0.030$) but not advanced prostate cancer (OR = 0.98, 95% CI = 0.89 to 1.07, $P = 0.600$) using the IVW method accounting for genetic correlation structure, consistent with findings from our initial analyses. Repeating the MR analysis using conditionally independent *cis*-pQTLs on overall PrCa risk provided a very similar magnitude of effect (OR = 0.92, 95% CI = 0.85 to 0.999, $P = 0.048$).

LDL-c removal occurs primarily in the liver, which is also the organ where *PCSK9* is strongly expressed based on the latest release (v8) of the Genotype-Tissue Expression (GTEx) project [54]. Based on our variant selection criteria (i.e., in and around the PCSK9 gene and pairwise $r^2 < 0.1$) and a false-discovery rate threshold of 0.05 defined by GTEx, there was only 1 eQTL in liver tissue using this dataset (rs553741), which is in strong LD with one of the SNPs used as a genetic instrument to proxy the effects of PCSK9 inhibition in the primary drug target MR (rs472495, $r^2 = 0.909$) (for full LD matrix see S9 Table). MR estimates were supportive of an association between lower levels of *PCSK9* gene expression (instrumented using the eQTL) and a lower risk of overall prostate cancer (OR = 0.90, 95% CI = 0.86 to 0.95, $P = 5.50 \times 10^{-5}$, per SD reduction in *PCSK9* transcript levels). Analysis on disease subtypes provided similar magnitude of association with genetically proxied PCSK9, although confidence intervals overlapped the null (early-onset prostate cancer: OR = 0.91, 95% CI = 0.91 to 1.03, $P = 0.139$; advanced prostate cancer: OR = 0.93, 95% CI = 0.86 to 1.01, $P = 0.102$).

As drug target MR analyses, particularly when using only a single *cis*-acting variant as an instrument, are susceptible to false-positive findings due to LD structure with nearby genes [55,56], we conducted genetic colocalization at this locus to evaluate evidence of a shared causal variant between *PCSK9* expression in liver tissue and prostate cancer. A LocusZoom plot comparing the *cis*-acting eQTLs associated with liver tissue-derived *PCSK9* expression and SNPs associated with risk of prostate cancer (Fig 3) identified a shared top SNP (rs553741) in the *PCSK9* gene region. Analysis using *coloc* method found a posterior probability of colocalization (H4) of 81.3%, providing strong evidence for colocalization between the 2 traits. Full results from *coloc* are presented in S10 Table. Using the eCAVIAR method [48], we found a CLPP of 0.103 for the variant rs553741, which suggests there is strong evidence of a shared causal variant at this locus based on a threshold of CLPP > 0.01 as proposed by the authors of this approach [48]. Detailed CLPP for every candidate SNP are presented in S11 Table.

In addition, LocusZoom plots for liver tissue-derived *HMGCR* and *NPC1L1* gene expression and prostate cancer did not find evidence supporting shared top hits in their respective gene regions (S3 and S4 Figs). Formal evaluations using *coloc* also found little evidence for colocalization between the expression of *HMGCR* or *NPC1L1* in liver and prostate cancer at the respective genes (H4 = 0.33% for *HMGCR*, H4 = 0.38% for *NPC1L1*). Additional LocusZoom plots visualising genetic variants associated with LDL-c and prostate cancer at *PCSK9*, *HMGCR*, and *NPC1L1* genes provided similar results (S5–S7 Figs). Genetic correlation results identified evidence for correlation between LDL-c and prostate cancer risk at the PCSK9 loci (correlation coefficient rho = 1) (S12 Table).

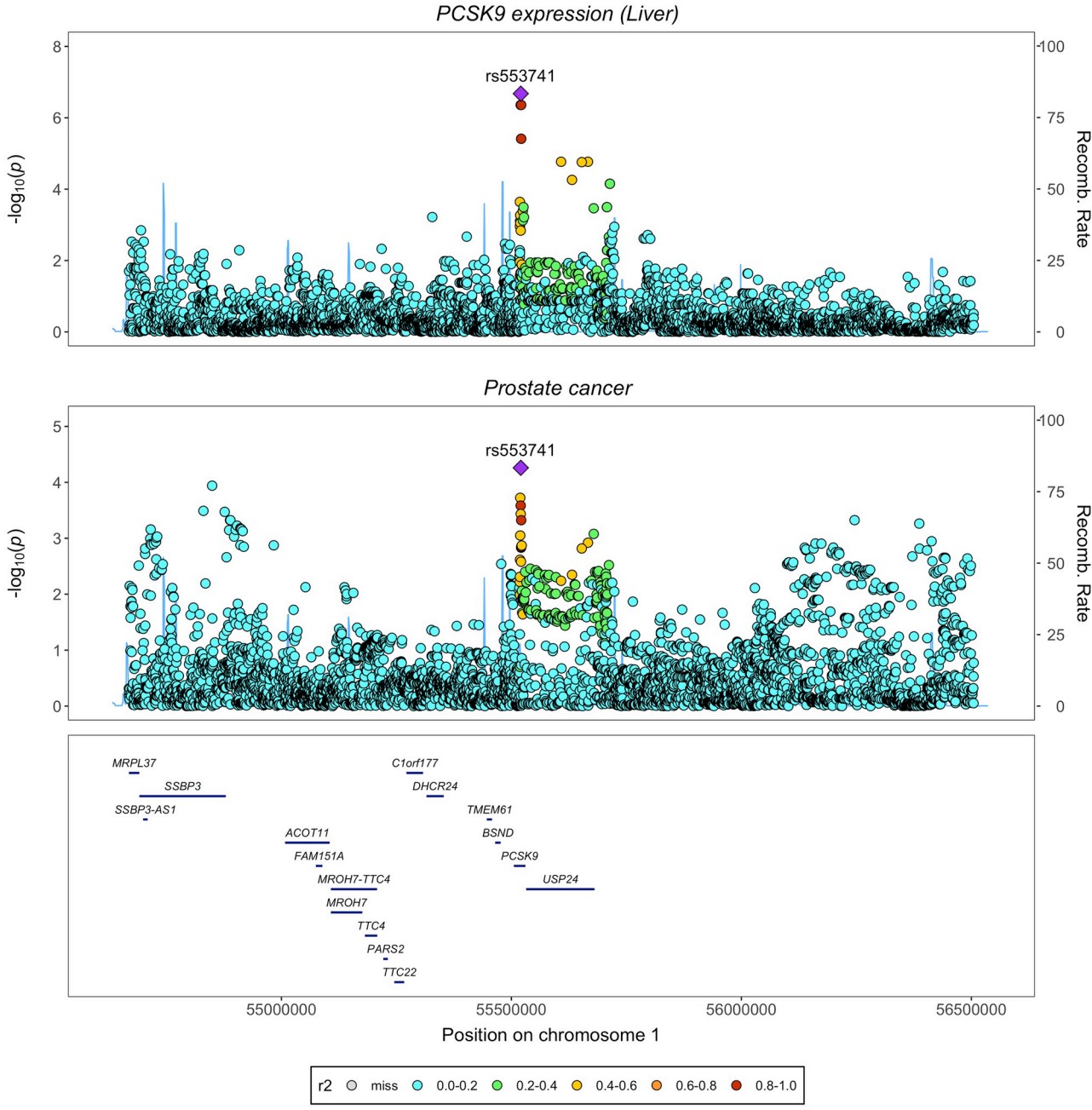

**Fig 3. LocusZoom plots illustrating evidence of genetic colocalization between *PCSK9* gene expression in the liver and prostate cancer risk at the *PCSK9* gene locus.**

## Contrasting the genetically proxied associations between lipid-lowering drug targets and risk factors of prostate cancer

We hypothesised that the association between genetically proxied lipid-lowering drug target inhibition and prostate cancer may be mediated through prostate cancer risk factors, such as BMI, Lp(a), or testosterone. Therefore, we examined the association between genetically proxied inhibition of drug targets and risk factors for prostate cancer using drug target MR (Fig 4).

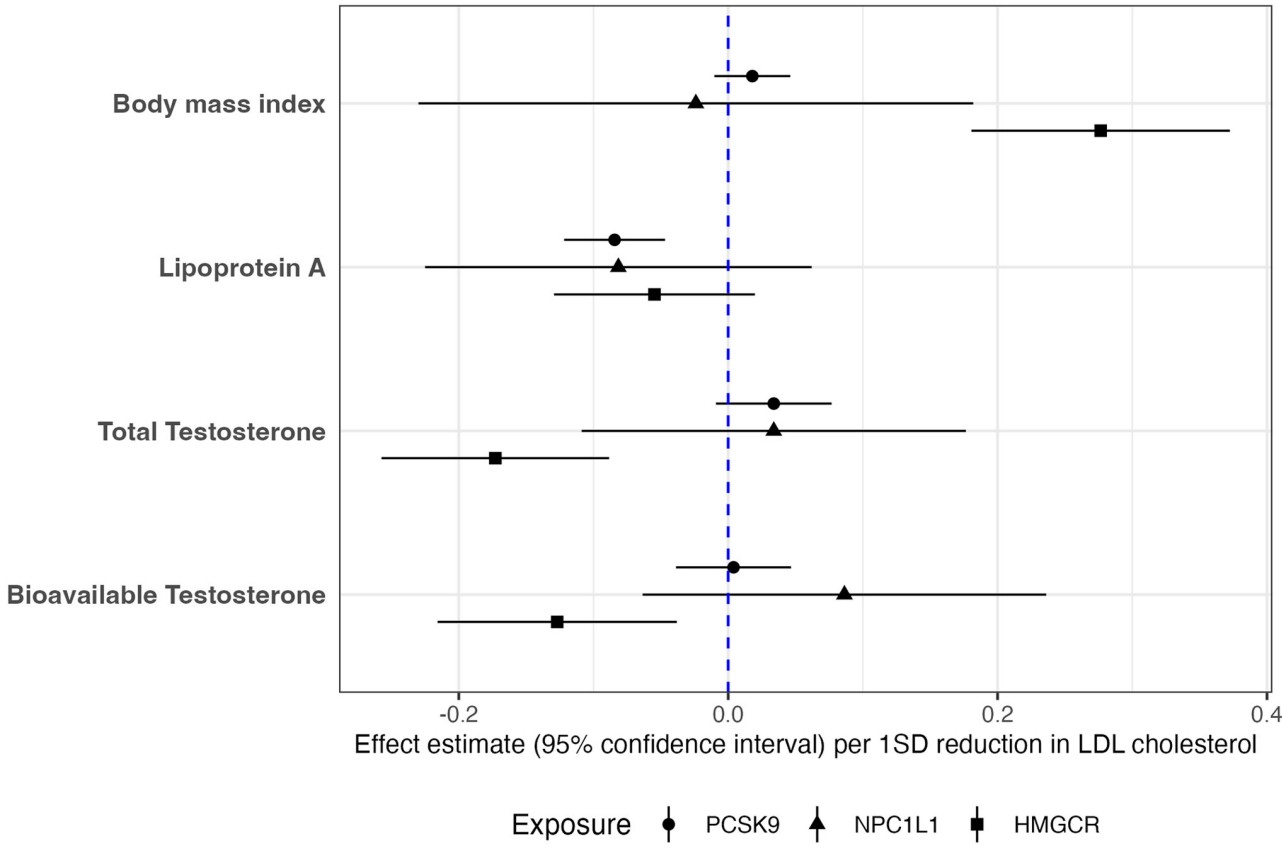

**Fig 4. Results from drug target MR analyses to investigate the effect of lipid-lowering therapies on BMI, lipoprotein A, and testosterone.** Estimates of effects from genetically proxied inhibition of PCSK9, NPC1L1, and HMGCR on BMI, levels of lipoprotein A, total testosterone, and bioavailable testosterone. Effect estimates were in SD change in the outcome per drug target inhibition effect equivalent to an SD reduction in LDL-c. Results were from analysis using instruments identified from the GLGC data. BMI, body mass index; GLGC, Global Lipids Genetics Consortium; LDL-c, LDL cholesterol; MR, Mendelian randomization; SD, standard deviation.

To maximise power, all effect estimates for BMI and Lp(a) reported in the main text are from analysis using males and females combined GWAS as the outcome. For validation analysis results using males only GWAS on BMI and Lp(a), see supporting files S13 and S14 Tables.

Repeating our primary MR analyses to investigate the genetically proxied association of each lipid-lowering target on BMI (S13 Table) provided little evidence for genetically proxied inhibition of PCSK9 (Beta = 0.02, 95% CI = −0.01 to 0.05, $P$ = 0.212, SD change in BMI per SD reduction in LDL-c) as well as NPC1L1 (Beta = −0.02, 95% CI = −0.23 to 0.18, $P$ = 0.819) on prostate cancer risk. However, genetically proxied inhibition of HMGCR provided strong evidence for an association with elevated BMI (Beta = 0.28, 95% CI = 0.18 to 0.37, $P$ = 1.61 × 10$^{-8}$). Replication using males-only GWAS on BMI provided similar evidence (S13 Table).

Evaluating effects on Lp(a) for each target (S14 Table) suggested that there was strong evidence of an effect of PCSK9 inhibition on lower levels of this lipoprotein particle (IVW accounting for LD matrix: Beta = −0.08, 95% CI = −0.12 to −0.05, $P$ = 1.00 × 10$^{-5}$, SD change in Lp(a) levels per SD reduction in LDL-c). The association was supported in an analysis using PCSK9 pQTLs as the genetic instruments (Beta = −0.03 SD change in the levels of Lp(a) per SD reduction in plasma PCSK9 levels, 95%CI = −0.05 to −0.02, $P$ = 1.47 × 10$^{-4}$). Conversely,

**Fig 5. A DAG showing the putative association between PCSK9 inhibition, lipoprotein A, and prostate cancer.** DAG, directed acyclic graph.

investigating the effects of genetically proxied inhibition of HMGCR (Beta = −0.05, 95% CI = −0.13 to 0.02, $P$ = 0.150) and NPC1L1 (Beta = 0.08, 95% CI = −0.44 to 0.02, $P$ = 0.080) on Lp (a) levels found that their CIs included the null despite similar central magnitudes of effect compared with PCSK9. Replication using males-only GWAS on Lp(a) provided similar evidence (S14 Table).

In addition, we examined the association between genetically proxied Lp(a) levels on prostate cancer risk from using male-stratified GWAS on Lp(a) by replicating MR analysis conducted by Ioannidou and colleagues [8]. Using 15 Lp(a)-associated variants from across the genome ($r^2 < 0.001$, $P < 5 \times 10^{-8}$), we found consistent effect estimates on the association between genetically proxied Lp(a) and prostate cancer in the univariable setting using IVW (OR = 1.06, 95% CI = 0.95 to 1.20, $P$ = 0.305, per SD increase in Lp(a) levels) and weighted median methods (OR = 1.07, 95% CI = 1.004 to 1.13, $P$ = 0.036). Analysis using the genetic variant associated with Lp(a) located within the *LPA* gene (rs73596816) provided evidence with a consistent magnitude of association (OR = 1.07, 95% CI = 1.00 to 1.14, $P$ = 0.056) based on the Wald ratio method. Multivariable MR using the *cis*-variant for Lp(a) adjusting for LDL-c levels provided strong evidence for genetically proxied Lp(a) on lower prostate cancer risk (OR = 1.05, 95% CI = 1.01 to 1.08, $P$ = 0.013). The putative causal relationship between PCSK9 inhibition, Lp(a), and prostate cancer risk is illustrated in a directed acyclic graph (Fig 5).

Examining the effects from drug targets on testosterone levels (S15 Table) suggest genetically proxied inhibition of both PCSK9 and NPC1L1 contributed very little to alterations in total testosterone (PCSK9: Beta = 0.04, 95% CI = −0.01 to 0.09, $P$ = 0.146, SD change in testosterone per SD reduction in LDL-c; NPC1L1: Beta = 0.02, 95% CI = −0.19 to 0.22, $P$ = 0.876) or bioavailable testosterone (PCSK9: Beta = −0.02, 95% CI = −0.07 to 0.03, $P$ = 0.341; NPC1L1: Beta = 0.08, 95% CI = −0.08 to 0.25, $P$ = 0.327) in men. On the contrary, genetic variants proxying the inhibition of HMGCR showed strong correlation with both measurements of testosterone (total: Beta = −0.21, 95% CI = −0.29 to −0.12, $P$ = $2.36 \times 10^{-6}$; bioavailable: Beta = −0.14, 95% CI = −0.25 to −0.03, $P$ = 0.014) in men. However, associations between genetically proxied HMGCR inhibition with testosterone and BMI require further evaluations, such as genetic colocalization analyses to investigate potential pleiotropic effects via neighbouring genes.

## Discussion

In this work, we have identified strong evidence using large-scale genetic data to suggest that therapeutic inhibition of lipid-lowering drug target PCSK9 may reduce prostate cancer risk. Estimates based on circulating PCSK9 protein and liver tissue-derived *PCSK9* expression data further support this finding. Taken together, these findings suggest that the genetically proxied association between PCSK9 inhibition and a lower risk of prostate cancer is unlikely to be due to a mechanism involving the lowering of LDL-c levels. We postulate that one potential

explanation for this finding is due to the lowering of Lp(a), which genetically proxied PCSK9 inhibition provided stronger evidence of achieving in comparison to statin and Ezetimibe therapies in this study.

PCSK9 is known as a regulator of the metabolism of LDL-c, although thus far its role in cancer susceptibility has yet to be comprehensively evaluated and characterised. There are various preclinical studies based on tumour tissue or mouse models that report the direct effects of PCSK9 or PCSK9 inhibitors on multiple types of cancer [57], including hepatocellular carcinoma [58,59], lung carcinoma [60,61], colorectal cancer [62], and breast cancer [63]. Additionally, Liu and colleagues inoculated *Pcsk9* knockout mouse cancer cells into syngeneic mouse hosts and observed delayed tumour growth, as well as a synergistic effect between PCSK9 inhibition and anti-PD1 antibody treatment to promote the efficacy of tumour growth suppression [64]. In contrast, few preclinical studies have linked PCSK9 to prostate cancer [65,66], although Gan and colleagues previously demonstrated that *PCSK9* siRNA protects human prostate cancer cells from ionising radiation-induced cell damage [67].

With excellent efficacy and safety profiles, 2 monoclonal antibodies against PCSK9 have been developed to lower elevated LDL-c levels and subsequently help prevent coronary heart disease (CHD) [68,69]. Our findings add to growing evidence suggesting that PCSK9 inhibitors may provide the most benefit towards reducing disease risk in comparison to statins and Ezetimibe that inhibit HMGCR and NPC1L1, respectively. Furthermore, recent evaluations involving CRISPR base editing in primates suggests that complete knockdown of PCSK9 in the liver results in approximately a 60% reduction of LDL cholesterol [70]. Further research is required to investigate the consequences of this approach towards prostate cancer risk, although our findings using genetic proxies of PCSK9 inhibition predict that this would have a beneficial effect, especially on early-onset prostate cancer. This supports the recently identified favourable overall effects from genetically proxied PCSK9 inhibition on the lifespan [71].

MR analyses provided a similar central effect estimate for genetically proxied HMGCR estimates and prostate cancer as those found for PCSK9 inhibition, although with wider corresponding confidence intervals resulting in weaker evidence of an effect. Furthermore, as was the case for NPC1L1, there was also weak evidence of a shared causal variant based on colocalization analyses. This could be due to the lack of power provided by the prostate cancer GWAS used in this study, meaning that investigations should be repeated in the future once findings from larger prostate cancer case-control GWAS are available.

The evidence for associations between genetically proxied inhibition of lipid-lowering drug targets and prostate cancer is consistent with a recently published drug target MR study [12], in which the authors applied a smaller set of genetic instruments for drug targets (11 for PCSK9, 3 for NPC1L1, and 5 for HMGCR) identified from the GWAS on LDL-c ($n$ = 173,082) published in 2013 [72] to study the effects of lipid-lowering drugs on prostate and breast cancer risk. In addition to the use of a larger set of genetic instruments identified from a much recent and larger GWAS meta-analysis, our study has provided a more comprehensive and robust evaluation. Firstly, we examined the association between on-target effects of lipid-lowering drugs and cancer risk by including genetically proxied LDLR and LDL-c as the exposures. Secondly, we conducted various sensitivity analyses to further investigate the finding between genetically proxied PCSK9 inhibition and prostate cancer risk. For example, leave-one-out analyses found consistent effect estimates for genetically proxied PCSK9 on prostate cancer risk, suggesting the association is unlikely to be driven by any single variant in our instrument. This mitigates the likelihood that individual pleiotropic variants at the *PCSK9* locus are influencing prostate cancer risk via alternate biological pathways. Thirdly, we have triangulated evidence from liver tissue-derived gene expression and plasma protein data to further support findings from our primary MR approach. Moreover, we included further

evidence from genetic colocalization analyses using liver tissue-derived gene expression data, which suggests that our findings are unlikely to be explained by a pathway involving a neighbouring gene as opposed to *PCSK9*. Furthermore, we included validation analyses using male-stratified datasets where possible.

Additionally, we performed subsequent analyses to explore the potential mediatory role of BMI, Lp(a), and testosterone in the associations between drug targets and prostate cancer risk. Elevated BMI has been found to associate with increased prostate cancer risk in multiple observational studies [26–28], whereas MR study identified evidence for an inversed association between them [73]. In addition, levels of testosterone were found to be strongly correlated with the risk of prostate cancer in both observational [29] and MR [30] studies. Our results found little evidence supporting effects of genetically proxied inhibition of PCSK9 on these 2 risk factors. Moreover, a higher levels of Lp(a) has previously been reported to associate with poor prognosis of prostate cancer [74] as well as a higher risk of overall and early-onset prostate cancer [8]. The evidence for associations between Lp(a) and overall prostate cancer risk was replicated in our study using male-stratified genetic instruments. Our results provided strong evidence that PCSK9 inhibition may lower levels of Lp(a), whereas effect estimates from the inhibition of NPC1L1 or HMGCR again had wider CIs compared to PCSK9 estimates. This finding has also been supported by randomised control trials [68,75] which found that PCSK9 inhibitors significantly reduced levels of Lp(a) (e.g., Alirocumab lead to 25.6% reduction in Lp(a) compared with placebo group at week 24 [68]), whereas statins and Ezetimibe had little to mild effects on Lp(a) [76]. Furthermore, genetically proxied PCSK9 exhibited the strongest magnitude of association with the risk of early-onset prostate cancer than that of overall prostate cancer, and there is weak evidence for association with advanced disease. This is consistent with the association between genetically predicted Lp(a) and the risk of prostate cancer outcomes identified in the recent multivariable MR [8] and replicated in our study. Although our findings suggest that Lp(a) may play a mediatory role along the pathway between PCSK9 inhibition and prostate cancer risk, further functional work is required to robustly demonstrate this.

This study has noteworthy limitations. Firstly, although the estimates derived in this MR study are based on a standard deviation change in LDL-c that is clinically achievable (e.g., in randomised controlled trials, PCSK9 inhibitors on average reduced LDL-c by 60 to 70 mg/dL compared with placebo group at week 24 in addition to the use of statins [68,69]), these are based on genetic proxies of therapeutic targets that may not be equivalent to those reported by randomised controlled trials. This is because drugs are often taken for a defined period, whereas estimates from MR analyses are conventionally interpreted as lifelong exposures to risk factors given that genetic variants are typically fixed at conception. Furthermore, there is increasing evidence to suggest that associations between genetic instruments and exposures may vary throughout the life course [77]. Additionally, conventional MR methods assume a linear relationship between genetically proxied exposures and outcomes; however, drugs may not trigger any biological response until a drug dose exceeds a certain level. Secondly, we leveraged 13 *cis*-pQTLs to instrument plasma levels of PCSK9 in this work due to the availability of a large-scale dataset for whole blood measures ($n$ = 35,559) [78]. However, analyses using *cis*-pQTL derived from liver tissue would be valuable to further examine the effect of PCSK9 inhibition on prostate cancer risk once these data are available in sufficient samples. Thirdly, using genetic instruments at the PCSK9 locus extracted from a GWAS of LDL-c, liver-derived PCSK9 expression, and circulating PCSK9 protein, this work focuses on the indirect association between PCSK9 inhibition and prostate cancer risk. Future studies with prostate derived PCSK9 *cis*-pQTL are essential for evaluating the direct role of PCSK9 in prostate cancer cells, especially the effect on advanced prostate cancer.

Additionally, replication of MR estimates using males-stratified PCSK9 *cis*-eQTL and *cis*-pQTL would be worthwhile to support this finding in the future, even though we did not find large differences using LDL-c stratified instruments to overall conclusions. We also note that multivariable MR cannot be applied to explore the mediatory role of Lp(a) due to the lack of genome-wide significant genetic variants in *PCSK9* gene region from the UK Biobank GWAS on Lp(a). It is worth further investigation when a larger GWAS on Lp(a) is available in the future. Moreover, the lack of evidence for an association between PCSK9 inhibition and the risk of advanced prostate cancer, as well as comparatively wider confidence intervals for associations between HMGCR, NPC1L1, and prostate cancer (compared to genetically proxied PCSK9 inhibition) could potentially be due to lack of power. These should be further explored when larger datasets are available. Furthermore, we did not explore the genetically proxied association between drug target inhibition and other stratified prostate cancer phenotypes, such as Gleason score, cancer aggressiveness, or recurrence. Although the PRACTICAL consortium has published GWASs on some of these phenotypes, these GWASs are conducted among prostate cancer cases without the inclusion of non-cancer controls. MR analysis using such datasets may induce index event bias [79], leading to false-negative results or even spurious inverse associations. These could be investigated by following up participants of randomised controlled trials for PCSK9 inhibitors. Similarly, leveraging large-scale summary-level data from PRACTICAL meant that we were unable to evaluate the time-varying effects of PCSK9 inhibition on risk of prostate cancer at separate stages in the life course or in population subgroups undergoing different treatment regimens. These are therefore important areas of future research to investigate either in a clinical trial setting or when large-scale genetic association datasets become accessible.

In summary, our study demonstrates that genetically proxied inhibition of PCSK9 is strongly associated with a lower risk of overall and early-onset prostate cancer, potentially through a mechanism involving the lowering of Lp(a) levels. Further evidence from clinical studies on prostate cancer incidence and progression among patients taking PCSK9 inhibitors is needed to confirm this finding.

## Supporting information

**S1 STROBE Checklist. Strengthening the Reporting of Observational Studies in Epidemiology using Mendelian Randomisation (STROBE-MR) checklist.**
(DOCX)

**S1 Supplementary Note. Data sources for secondary analyses.**
(DOCX)

**S1 Table. Details of GWAS summary statistics involved in this study.**
(XLSX)

**S2 Table. Genetic variants used as instrumental variables for drug targets and LDL cholesterol levels in Mendelian randomisation analysis, identified from GLGC GWAS meta-analysis on LDL cholesterol.**
(XLSX)

**S3 Table. Genetic variants used as instrumental variables for drug targets and LDL cholesterol levels in Mendelian randomisation analysis, identified from UK Biobank GWAS on LDL cholesterol.**
(XLSX)

**S4 Table. PCSK9 liver eQTLs and plasma pQTL used as instrumental variables in secondary MR analysis.**
(XLSX)

**S5 Table. Functional annotations of LDL-c associated genetic variants at the *PCSK9* locus.**
(XLSX)

**S6 Table. Results from Mendelian randomisation investigating the associations between genetically proxied inhibition of drug targets, reduction of LDLR levels of LDL-c levels, and the risk of prostate cancer.** Estimates are effects equivalent to 1 SD reduction in LDL-c levels.
(XLSX)

**S7 Table. Results from sensitivity tests on the PCSK9 drug target Mendelian randomisation on total prostate cancer risk.** These analyses were performed using genetic instruments identified from the Global Lipids Genetics Consortium.
(XLSX)

**S8 Table. Results from Mendelian randomisation investigating the associations between plasma PCSK9 pQTL and liver-derived PCSK9 eQTL and prostate cancer outcomes.** Estimates are per 1 SD reduction in plasma protein levels of PCSK9 for pQTLs or per 1 SD reduction in the levels of *PCSK9* transcripts in the liver tissue.
(XLSX)

**S9 Table. Pairwise linkage disequilibrium correlation coefficients between liver-derived PCSK9 *cis*-eQTL and plasma PCSK9 pQTLs.** The pairwise $r^2$ were calculated using the reference panel consisting of Utah Residents from North and West Europe (CEU) individuals from the 1,000 Genomes project.
(XLSX)

**S10 Table. Results from colocalization analysis using *coloc* methods.**
(XLSX)

**S11 Table. Colocalization posterior probability of candidate causal genetic variants shared between PCSK9 gene expression in liver and prostate cancer risk.**
(XLSX)

**S12 Table. Results from genetic correlation analysis using LD Score regression and LAVA.**
(XLSX)

**S13 Table. Results from Mendelian randomisation analysis on the associations between genetically proxied inhibition of drug targets, reduction of LDLR levels of LDL-c levels, and body mass index.** Estimates are effects equivalent to 1 SD reduction in LDL-c levels or 1 SD reduction in plasma protein levels of PCSK9.
(XLSX)

**S14 Table. Results from Mendelian randomisation analysis on the associations between genetically proxied inhibition of drug targets, reduction of LDLR levels of LDL-c levels, and the levels of lipoprotein A.** Estimates are effects equivalent to 1 SD reduction in LDL-c levels or 1 SD reduction in plasma protein levels of PCSK9.
(XLSX)

**S15 Table. Results from Mendelian randomisation analysis on the associations between genetically proxied inhibition of drug targets, reduction of LDLR levels of LDL-c levels,**

**and the levels of testosterone.** Estimates are effects equivalent to 1 SD reduction in LDL-c levels or 1 SD reduction in plasma protein levels of PCSK9.
(XLSX)

**S1 Fig. Regulatory associations between all genes involved in this study.**
(TIFF)

**S2 Fig. Forest plot showing the results from leave-one-out Mendelian randomisation analyses of associations between PCSK9 inhibition and prostate cancer risk.** SD, standard deviation.
(TIFF)

**S3 Fig. LocusZoom plots comparing genetic variants associated with *HMGCR* gene expression in the liver and prostate cancer risk at the *HMGCR* gene locus.**
(TIFF)

**S4 Fig. LocusZoom plots comparing genetic variants associated with *NPC1L1* gene expression in the liver and prostate cancer risk at the *NPC1L1* gene locus.**
(TIFF)

**S5 Fig. LocusZoom plots comparing genetic variants associated with LDL cholesterol and prostate cancer risk at the *PCSK9* gene locus.**
(TIFF)

**S6 Fig. LocusZoom plots comparing genetic variants associated with LDL cholesterol and prostate cancer risk at the *HMGCR* gene locus.**
(TIFF)

**S7 Fig. LocusZoom plots comparing genetic variants associated with LDL cholesterol and prostate cancer risk at the *NPC1L1* gene locus.**
(TIFF)

**S1 File. List of PIs from the PRACTICAL (http://practical.icr.ac.uk/), CRUK, BPC3, CAPS, PEGASUS consortia, and extended acknowledgement for these consortia.**
(DOCX)

## Author Contributions

**Conceptualization:** Si Fang, Tom G. Richardson.

**Data curation:** Si Fang, Tom G. Richardson.

**Formal analysis:** Si Fang.

**Investigation:** Si Fang, Tom G. Richardson.

**Methodology:** Si Fang, James Yarmolinsky, Dipender Gill, Tom G. Richardson.

**Resources:** Caroline J. Bull, Claire M. Perks, Tom G. Richardson.

**Software:** Si Fang, Tom G. Richardson.

**Supervision:** George Davey Smith, Tom R. Gaunt, Tom G. Richardson.

**Visualization:** Si Fang.

**Writing – original draft:** Si Fang, Tom G. Richardson.

**Writing – review & editing:** Si Fang, James Yarmolinsky, Dipender Gill, Caroline J. Bull, Claire M. Perks, George Davey Smith, Tom R. Gaunt, Tom G. Richardson.

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
