## [Editor Report · Decision Letter 0]

14 Apr 2022

Dear Dr Fang, 

Thank you for submitting your manuscript entitled "Genetically proxied PCSK9 inhibition provides indication of lower prostate cancer risk: a Mendelian randomization study" for consideration by PLOS Medicine.

Your manuscript has now been evaluated by the PLOS Medicine editorial staff and I am writing to let you know that we would like to send your submission out for external assessment.

However, we first need you to complete your submission by providing the metadata that is required for full assessment. To this end, please login to Editorial Manager where you will find the paper in the 'Submissions Needing Revisions' folder on your homepage. Please click 'Revise Submission' from the Action Links and complete all additional questions in the submission questionnaire.

Please re-submit your manuscript within two working days, i.e. by Apr 18 2022 11:59PM.

Once your full submission is complete, your paper will undergo a series of checks in preparation for assessment

Kind regards,

Richard Turner, PhD

rturner@plos.org

---

## [Decision Letter · Decision Letter 1]

26 Jul 2022

Dear Dr. Fang,

Thank you very much for submitting your manuscript "Genetically proxied PCSK9 inhibition provides indication of lower prostate cancer risk: a Mendelian randomization study" (PMEDICINE-D-22-01177R1) for consideration at PLOS Medicine. 

[LINK]

In light of these reviews, I am afraid that we will not be able to accept the manuscript for publication in the journal in its current form, but we would like to consider a revised version that addresses the reviewers' and editors' comments. Obviously we cannot make any decision about publication until we have seen the revised manuscript and your response, and we plan to seek re-review by one or more of the reviewers. 

We hope to receive your revised manuscript by Aug 16 2022 11:59PM. Please email us (plosmedicine@plos.org) if you have any questions or concerns.

We look forward to receiving your revised manuscript. 

Sincerely,

Callam Davidson, 

Associate Editor

PLOS Medicine

plosmedicine.org

Comments from the Academic Editor:

1) Genetic variants used to construct the instruments were derived from a paper published in 2013. It might be worthwhile to search for more recent papers to see if additional variants have been identified for these targets;

2) The authors may want to perform a MRA to directly assess the association of Lp(a) with prostate cancer risk;

3) it would be helpful to mention in the abstract that the association with PCSK9 is not mediated through reducing LDL-C;

4) It is interesting to see the association between HMGCR and testosterone, which is a major risk factor for prostate cancer. However, no association was identified between HMGCR and prostate cancer risk. The authors may want to discuss this finding.

Please revise your title according to PLOS Medicine's style. Your title must be nondeclarative. It should begin with main concept if possible. For example: "Association between genetically proxied PCSK9 inhibition and prostate cancer risk: a Mendelian randomization study", or similar.

Please add this statement to the manuscript's Competing Interests: "GDS is an Academic Editor on PLOS Medicine's editorial board."

Abstract Methods and Findings:

* Please ensure that all numbers presented in the abstract are present and identical to numbers presented in the main manuscript text.

Please ensure that the study is reported according to the STROBE-MR guideline, and include the completed STROBE-MR checklist as Supporting Information. Please add the following statement, or similar, to the Methods: "This study is reported as per the Strengthening the Reporting of Observational Studies in Epidemiology (STROBE) guideline, specific for mendelian randomization (S1 Checklist)."

The STROBE-MR guideline can be found here: https://www.strobe-mr.org/

Please delete the competing interests information from references #11,#27, and #66. 

Comments from the reviewers:

Reviewer #1: The authors present an intriguing drug target MR study to evaluate the benefit of cholesterol lowering drugs to reduce the risk of prostate cancer. Overall, the study is well-conducted, and the analysis is largely convincing. I have a few comments to improve the presentation and suggestions for additional analysis.

1. Data sources: It would be fantastic if the authors could update the genetic associations with LDL-c to use the latest GLGC release instead of the 2013 one.

2. Data sources: cis-pQTLs genetic associations were estimate in the Icelandic population, yet clumping was performed using a European ancestry panel. How transferable is the European ancestry to the Icelandic ancestry? Could the authors find a better matching reference panel for LD estimation?

3. Co-localization between eQTL and total prostate cancer risk: Which genetic correlation matrix was used for the calculation of eCaviar? 

4. Co-localization between eQTL and total prostate cancer risk: Could the authors complement the eCaviar results with the more commonly used colocalisation package coloc or coloc-susie? Looking at the regional Manhattan plot there is only one causal variant in the region, so even standard colocalisation should be able to pick up the signal.

5. Methods: Lipid lowering drugs targets and risk factors of prostate cancer: Would it be possible to use only male-specific summary-level associations for the risk factors (BMI, Lipoprotein (a)) investigated?

6. Results: Lipid lowering drug targets and risk factors of prostate cancer

This is an extremely interesting section, yet the author could go one step further. The current analysis only estimates the arrow from Exposure to Mediator in Figure 4. The quantity of interest would be the percentage of effect that is mediated via the prostate cancer risk factors. Could the authors please include a summary-level mediation MR analysis (1) as well or are there too few instruments to obtain reliable estimates?

7. Discussion: The authors state: "The lack of evidence for an association between PCSK9 inhibition and the risk of advanced prostate cancer suggest circulating PCSK9 is less likely to be promoting the progression of the disease." Could the authors please perform a power calculation to complement this statement to demonstrate that the lack of evidence is not due to a lack of power. 

8. General: A recent two-sample MR study has shown a potential causal role for Lipoprotein (a), yet very little evidence for LDL-c (2). Could the authors repeat their drug-target MR analysis using genetic associations with Lipoprotein (a) instead of genetic associations with LDL-c?

Minor: 

- "Contrasting the genetically proxied associations between lipid lowering drugs targets and risk factors of prostate cancer" while in the Results it is "Contrasting the genetically proxied associations between lipid lowering drug targets and risk factors of prostate cancer". It would be good to harmonize to "drug targets".

References

1. D. Gill et al., Int J Obes (Lond). 45, 1428-1438 (2021).

2. A. Ioannidou et al., PLOS Medicine. 19, e1003859- (2022).

Reviewer #2: 

The authors conducted a drug-target Mendelian randomisation analysis (MR) to examine the potential effect of targetting PCSK9, HMGCR, NPC1L1, and LDLR for LDL-C reduction on the risk of prostate cancer. The idea is scientifically and clinically sound and robust. The finding does not directly impact clinical practice but should interest readers of a clinical journal. However, the novelty is weak as follows.

A manuscript with the same idea has been available as follows:

Sun et al., Associations of genetically proxied inhibition of HMG-CoA reductase, NPC1L1, and PCSK9 with breast cancer and prostate cancer, Breast Cancer Res, 2022 Feb. doi: 10.1186/s13058-022-01508-0

Sun et al. conducted MR similar to the primary analysis of this submitted manuscript. The primary data sources were the same: quantitative genetic results from the GLGC and the PRACTICAL consortium. 

Sun et al. reported an odds ratio of 0.81 (95% CI 0.73-0.90) per 1 standard deviation lower LDL-c predicted with the genetic instrument specifically for PCSK9-related loci. The authors of this submitted manuscript reported 0.84 (0.74-0.96). The same non-significant results were observed in both studies for the other sets of gene variants related to LDL-c. Two studies took different ways to select loci within and near each of the PCSK9 gene and the others. Other methodological differences were present. The current study evaluated UK BioBank data, too, to examine male-specific genetic effects and support reproducibility. Mechanistic analyses and results are available, for example, with the foci on PCSK9 expression and Lp(a). Specific strengths were terrific compared to the study by Sun et al. 

However, the authors neither documented nor discussed the limited novelty appropriately, overstating the novelty. The authors should cite the paper by Sun et al. several times in this manuscript. The introduction section should indicate the availability and implication of the study by Sun et al. Then, the authors should justify why this study is essential for the clinical audience, given the previous effort by Sun et al. In the discussion section, the authors should document what additional evidence they could provide above and beyond that from Sun et al.

Minor comments:

The introduction should clarify what MR evidence was available before this study. The publications of #5 and #27, for example, seem to be the prior MR partly promising LDL-C lowering for lowering the risk of prostate cancer. The introduction should clarify what evidence was available from MR or non-genetic observational studies. Otherwise, readers would gain impression that the authors are hiding prior MR evidence about lipids and prostate cancer.

The authors may better provide Miami plots for GWAS results for LDL-c and prostate cancer for each genetic region of interest (PCSK9, HMGCR, etc.). Figure 1 visually indicates the genetic correlations between genetic effects on LDL-C and those on prostate cancer. It would be nice to see the same figures for the other genes in the supplementary materials: those may not visually indicate any genetic correlations or one promising for HMGCR.

The authors may better quantify genetic correlations without selecting specific loci for each gene with user-defined criteria. The authors may do LD-score regression as secondary analysis, subsetting the genetic data to a particular genetic region.

Discussion:

The authors should discuss the clinical pharmacological effect of lowering LDL-c or Lp(a) was comparable with the estimate from MR. The authors have documented the potential lack of the comparability as a limitation. However, they can collectively and quantitatively discuss it based on their estimate and published information. 

Other minor comments are reserved at this moment.

Reviewer #3: The authors present an analysis of strong association of the genetically proxied inhibition of PCSK9 with a lower risk of overall and early onset prostate cancer, potentially through a mechanism involving the lowering of Lp(a) levels. The authors explored a very interesting and clinically relevant question utilizing previously collected genetic sequencing and phenotype data on a cohort of men in UK Biobank study.

Major Comments:

1. Authors should provide a table on Summary of Demographics and Clinical Characteristics of the study cohort

2. Authors should describe in methods: how they defined the advanced prostate cancer and controls in specific groups (early onset and advanced PCa group) and if the case and controls are matched (eg. age, serum PSA, Family history, clinical follow-up time)? Was the analysis adjusted to significant confounding variables?

3. Authors should provide functional evidence of how the variants affects the gene and regulatory pathway for lipoprotein metabolism in context of PCa.

Minor Comments:

1. How the association for genetically proxied inhibition of PCSK9 stands, when the advanced PCa group is split in subgroups (eg. metastasis group, high Gleason grade group etc.). The results may be better refined since advanced PCa considered in the study is heterogenous.

2. Since the treatment regimen and age can affect the levels of hormones and lipids in circulation, authors may want to comment on these aspects and how they incorporated this variability in the present study 

3. Authors may include analysis on biochemical recurrence, if clinical follow up on patients is available.

[LINK]

---

## [Decision Letter · Decision Letter 2]

2 Nov 2022

Dear Dr. Fang,

Thank you very much for re-submitting your manuscript "Association between genetically proxied PCSK9 inhibition and prostate cancer risk: a Mendelian randomization study" (PMEDICINE-D-22-01177R2) for review by PLOS Medicine.

I have discussed the paper with my colleagues and the academic editor and it was also seen again by three reviewers. I am pleased to say that provided the remaining editorial and production issues are dealt with we are planning to accept the paper for publication in the journal.

[LINK]

We look forward to receiving the revised manuscript by Nov 09 2022 11:59PM.   

Sincerely,

Callam Davidson, 

Associate Editor 

PLOS Medicine

plosmedicine.org

Requests from Editors:

The Data Availability Statement (DAS) requires revision. If data are freely available upon request, please state the owner of the data set and contact information for data requests (web or email address). Note that a study author cannot be the contact person for the data.

The ‘Author Contributions’, ‘Funding’, and ‘Conflicts of Interest’ sections can all be removed from the main manuscript, but it is essential that this information is all captured via your responses to the submission form – when submitting your revision, please ensure all information is included in your answers to the relevant questions. 

Please include continuous line numbering throughout your manuscript. 

Please ensure that all numbers presented in the abstract are present and identical to numbers presented in the main manuscript text.

To help us extend the reach of your research, please provide any Twitter handle(s) that would be appropriate to tag, including your own, your coauthors’, your institution, funder, or lab. Please respond to this email with any handles you wish to be included when we tweet this paper.

Comments from Reviewers:

Reviewer #1: The authors have addressed all of my comments.

Reviewer #2: The authors revised the manuscript reasonably well. The Reviewer has come up with minor comments, but the revision of the introduction should be made clearly accordingly.

Abstract:

Methods and findings: The data use should be more explicit. The authors should clarify whether they meant the use of publically available summary statistics or new in-house or collaborative analyses. "SNPs were used" and "Association estimates were obtained" are unclear. In recent MR literature, using summary statistics only seems to be common. If the authors did otherwise, e.g. sex-specific GWAS for lipids, they may want to clarify it. The authors may wish to clarify it in Figure 1, too.

The introduction section on page 6 requires some re-organisation. The authors explain MR analysis in the second paragraph but introduce the existing evidence based on MR in the first paragraph.

Page 6 should explicitly include the information on previous MR analyses that linked lipid-lowering loci of the specific genes to prostate cancer risk. 

In the second paragraph, the authors should cite the ref #12, present the previous results from the most relevant MR analysis, and explain why this study was necessary over and above what Ref 12 showed. As previously commented, the MR analysis conducted by Sun et al. (Ref 12) has already highlighted the idea of the drug-targetted MR for prostate cancer research. The authors should not introduce the approach as if they did it first in the context of lipid-lowering drugs and prostate cancer prevention. It would be expected to see a sentence starting with "Sun et al. did XXX. They reported XXX."

On Page 8, "Our work involves" should be in the past tense.

Figure 2B is better to put in the supplementary material. Compared to Figure 2A, 2B is not necessary, just supporting the robustness of the PCSK9 result.

Figure 2A should have a greater information on the left label. The gene information may better be supplemented with the number of SNPs and F statistics, for example.

The legend of Figure 2 should include the exposure unit: e.g. genetically estimated 1-SD increase in LDL-cholesterol in the GLGC.

Figure 4 legend should say "Effect estimates were in..." as the exposure unit the authors used in their analysis.

Reviewer #3: The authors have very well addressed my comments and critiques.

[LINK]

---

## [Editor Report · Decision Letter 3]

18 Nov 2022

Dear Dr Fang, 

On behalf of my colleagues and the Academic Editor, Dr Wei Zheng, I am pleased to inform you that we have agreed to publish your manuscript "Association between genetically proxied PCSK9 inhibition and prostate cancer risk: a Mendelian randomization study" (PMEDICINE-D-22-01177R3) in PLOS Medicine.

PRESS

Sincerely, 

Callam Davidson 

Associate Editor 

PLOS Medicine